# Structural basis of Ty3 retrotransposon integration at RNA Polymerase III-transcribed genes

Guillermo Abascal-Palacios [1✉], Laura Jochem[1], Carlos Pla-Prats[2], Fabienne Beuron [1] & Alessandro Vannini [1,3✉]

Retrotransposons are endogenous elements that have the ability to mobilise their DNA between different locations in the host genome. The Ty3 retrotransposon integrates with an exquisite specificity in a narrow window upstream of RNA Polymerase (Pol) III-transcribed genes, representing a paradigm for harmless targeted integration. Here we present the cryo-EM reconstruction at 4.0 Å of an active Ty3 strand transfer complex bound to TFIIIB transcription factor and a tRNA gene. The structure unravels the molecular mechanisms underlying Ty3 targeting specificity at Pol III-transcribed genes and sheds light into the architecture of retrotransposon machinery during integration. Ty3 intasome contacts a region of TBP, a subunit of TFIIIB, which is blocked by NC2 transcription regulator in RNA Pol II-transcribed genes. A newly-identified chromodomain on Ty3 integrase interacts with TFIIIB and the tRNA gene, defining with extreme precision the integration site position.

[1] Division of Structural Biology, The Institute of Cancer Research, London SW7 3RP, UK. [2] Friedrich Miescher Institute for Biomedical Research, Basel, Switzerland. [3] Human Technopole, 20157 Milan, Italy. ✉email: guiller.abascalpalacios@icr.ac.uk; alessandro.vannini@fht.org

Transposable elements (TEs) are mobile DNA sequences that can affect integrity and stability of host genomes, potentially disrupting genetic information and providing sites for homologous recombination. Thus, not surprisingly, TEs are implicated in several human diseases[1,2]. Long-terminal repeat (LTR) retrotransposons are a class of TEs that constitute a significant fraction of eukaryotic genomes and their "copy-and-paste" mobilisation into new loci occurs via an RNA intermediate, relying on the intrinsic reverse transcriptase (RT) and integrase (IN) activities[3–5].

Ty3/Gypsy is a member of the *Metaviridae* family of LTR-retrotransposons characterised by a highly defined targeting pattern in *S. cerevisiae*, integrating in a narrow location 2–3 bp upstream of RNA Pol III-transcribed genes[6,7]. Ty3/Gypsy life cycle is similar to retroviruses but it is confined within a cell. Tethering of transposable elements and retroviruses to host factors, in order to integrate at specific genomic location, is not confined only to Ty3/Gypsy TEs[8–11]. For example, Ty1/Copia retrotransposon interacts directly with subunits of RNA Pol III[12–15], *S. pombe* Tf1 element interacts with the Atf1p protein[8] and the human immunodeficiency virus (HIV) integrase binds to the transcriptional co-activator lens epithelium-derived growth factor (LEDGF/p75)[16–18].

RNA Polymerase (Pol) III is responsible for the synthesis of short untranslated RNAs such as the U6 spliceosomal RNA or the transfer RNAs (tRNAs)[19]. The recruitment to its target genes depends on a set of general transcription factors (GTFs) that mediates very stable binding to specific DNA promoter elements[20]. TFIIIB, which is minimally required for initiation of Pol III transcription[21–25], binds upstream of the transcription start site (TSS)[26] and consists of the TATA-box binding protein (TBP), the B-related factor 1 (Brf1)[27] and the B double prime (Bdp1) subunit[28]. TFIIIC, a 6-subunit GTF, recognises specific Pol III intragenic promoter elements and is required for the recruitment of TFIIIB at TATA-less promoters[29].

The minimal factors required for Ty3 targeting at Pol III genes are the TFIIIB subunits, Brf1 and TBP, whereas TFIIIC is essential for directionality of integration, which occurs in a narrow window of 2–3 base pairs (bp) upstream the transcriptional start site (TSS)[30–32]. Direct interactions between TFIIIB and the Ty3 IN have been reported in the past but the molecular mechanisms by which such an exquisite specificity is achieved remain elusive. Ty3 represents a paradigmatic model to study retrotransposon's targeted integration in host genomes. Notably, recent studies suggest that reactivation of endogenous retroelements in *H. sapiens*, a vertebrate endogenous retrovirus closely related to Ty3, is linked to disease development[33,34], including neurological disorders[35,36] and cancer[37,38]. Despite recent advances[39,40], the structural analysis of host-retroelement targeting has been restricted to cryo-electron-microscopy (cryo-EM) structures of histone-integrase interactions in nucleosomal context[41,42] and to crystal structures of HIV integrase domains bound to LEDGF/p75[16].

Here, we report the reconstitution of the Ty3 integration machinery (hereinafter referred as intasome), the biochemical characterisation of the Ty3 intasome interaction with the host factors TFIIIB and TFIIIC and, finally, the 4.0 Å-resolution cryo-EM reconstruction of a Ty3 intasome targeting and integrating at a TFIIIB-bound tRNA gene promoter.

## Results

**In vitro reconstitution of an active Ty3 retrotransposon machinery.** To obtain insights into the molecular mechanisms of Ty3 retrotransposition, we reconstituted the Ty3 intasome in vitro using recombinant Ty3 integrase (Supplementary Fig. 1a) and Cy5-labelled oligonucleotides corresponding to the Ty3 gene LTR termini (Fig. 1a and Supplementary Fig. 1b, c). In order to probe target-specific integration of the reconstituted Ty3 intasome, Ty3 activity was assessed by monitoring the specific recruitment to RNA Pol III transcription factors bound to a 157 bp-long synthetic tD(GUC)K gene promoter followed by the identification of the DNA products resulting from integration events (Fig. 1).

After step-wise binding of the TFIIIC and TFIIIB transcription factors to the tD(GUC)K gene promoter (Fig. 1b, lanes 2–4), a pre-assembled Ty3 intasome was added to the reaction. Electrophoretic mobility shift assays (EMSA) confirmed the ability of the reconstituted Ty3 intasome to specifically bind the tD(GUC)K gene promoter (Fig. 1b, lanes 5–6 and Supplementary Fig. 1d). The binding of the Ty3 intasome relies on the presence of the TFIIIB-TFIIIC general transcription factor complex as no binding was detected to the 157 bp FITC-labelled target DNA in their absence (Fig. 1b, lane 5 vs lane 8).

Evaluation of the integration activity was carried out following as a proxy the cleavage products of the tD(GUC)K nucleic acid scaffold, which would be expected from an integrative event by the in vitro Ty3 intasome, which is reconstituted with two linear short TR motifs as opposed to a single linear Ty3 gene with LTR termini (Fig. 1c, d).

Taking into consideration that the TSS is asymmetrically located in the target DNA (Fig. 1a), predicted Ty3 integration 2–3 bp upstream the TSS would result in two DNA fragments (hereinafter, upstream and downstream fragments) of ~75 and 120 bp length, respectively (Fig. 1a, inset). To identify the DNA products, the integration reaction was performed on target DNAs labelled with FITC fluorophore at the downstream end and a Ty3 intasome assembled with Cy5-labelled LTRs (Fig. 1c). The presence of a ~120 bp band labelled with FITC and Cy5 confirmed the integration event and was only observed upon addition of Ty3 intasome (Fig. 1c, lane 2 vs lane 3, fragment II). To further verify the identity of this DNA fragment, the reaction was repeated using an unlabelled intasome, which provided a FITC-only band of a similar length (Fig. 1c, lane 5, fragment II*), confirming this as the downstream integration product. Similarly, a ~75 bp Cy5-labelled band was also observed after Ty3 intasome addition, which was suggestive of an upstream integration product (Fig. 1c, lane 2 vs lane 3, fragment III). To confirm this hypothesis, a similar experiment was performed using a target DNA labelled only with a Cy3 fluorophore at the upstream end (Fig. 1d). This reaction produced a Cy3-labelled band of ~75 bp, which is only observed in the presence of Ty3 intasome (Fig. 1d, lane 2 vs lane 3, fragment III*), confirming this fragment as the upstream product of Ty3 integration. Reactions performed in the absence of TFIIIB and TFIIIC did not generate any integration products (Fig. 1c, lane 3 vs lane 4; and Fig. 1d, lane 3 vs lane 4), which confirmed the requirement of the transcription factors for Ty3 intasome recruitment to RNA Pol III-transcribed genes and the specificity of our in vitro assay. Lastly, analysis of the fluorescent products revealed also the presence of 157 bp non-integrated target DNA (Fig. 1c, fragment I; and Fig. 1d, fragment I*), which suggested that the in vitro reaction was not 100% efficient. Furthermore, in order to probe the role of TFIIIC in Ty3 integration, equivalent experiments were carried out in absence of TFIIIC (Supplementary Fig. 1e, f). In this context, TFIIIB alone was sufficient to efficiently recruit Ty3 intasome resulting in integration events qualitatively indistinguishable from the ones observed in presence of TFIIIC but with a reduced efficiency.

Overall, these results confirm that Ty3 in vitro reconstitution resulted in an active Ty3 retrotransposon machinery, displaying TFIIIB-dependent integration specificity that recapitulates Ty3 transposition at RNA Pol III-transcribed genes observed in vivo.

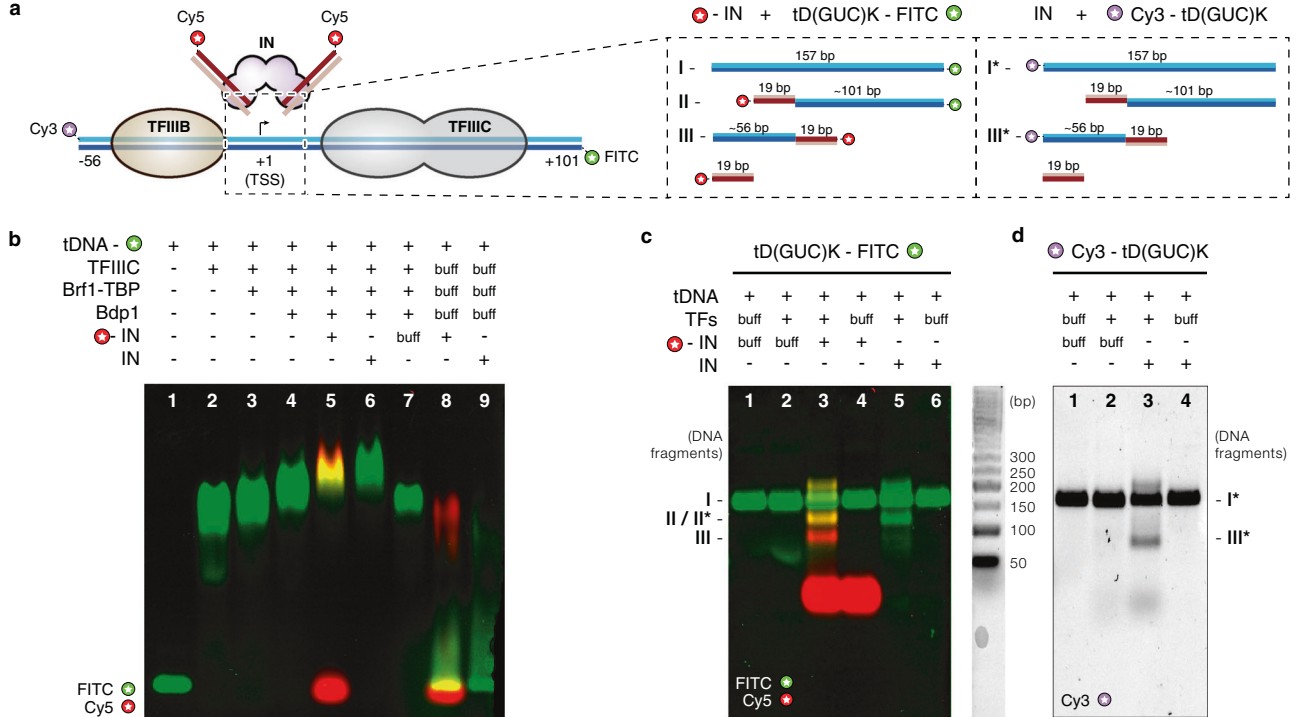

**Fig. 1 Assembly and activity of Ty3 retrotransposon machinery. a** Schematic representation of components used during binding and activity assays with reconstituted Ty3 retrotransposon machinery. *tD(GUC)K* template and non-template DNA strands are depicted as dark and light blue lines, respectively. DNA is numbered relative to the transcription start site (TSS). Reactive and non-reactive DNA strands of Ty3 LTRs are shown as dark and light red lines, respectively. TFIIIB and TFIIIC transcription factors, and intasome (IN) complex are depicted as globular shapes and coloured in wheat, grey and pink, respectively. Position and detail of the fluorophores used in the assays are shown as starred circles. The same colour scheme is used for fluorophores in **b**, **c**. Inset, summary and estimated length of the DNA products expected from integration events in the proximity of the TSS. **b** Analysis of complex assembly. Electrophoretic mobility shift assay (EMSA) shows sequential binding of transcription factors to the *tD(GUC)K* DNA (lanes 1–4), followed by binding of either Cy5-labelled Ty3 intasome (lane 5) or unlabelled intasome (lane 6) or intasome buffer (lane 7). Incubation of Cy5-labelled Ty3 intasome (lane 8) or unlabelled intasome (lane 9) with *tD(GUC)K* template in the absence of transcription factors does not lead to a recruitment to the DNA. **c** Integration activity assays of reconstituted Ty3 intasome (IN) into FITC-labelled *tD(GUC)K* tDNA or **d** Cy3-labelled *tD(GUC)K* tDNA, in the presence or absence of pre-assembled TFIIIC and TFIIIB transcription factors (TFs). DNA products were analysed in 4.5% agarose gels after proteinase K treatment. Insets specify fluorophore detection (FITC, Cy3 or Cy5). Integration activity assays shown in **c, d** were repeated in triplicate.

**Cryo-EM structure of Ty3 strand-transfer complex**. To gain structural insights into the molecular mechanisms of Ty3 retro-transposon integration at RNA Pol III-transcribed genes, we purified the full and minimal integration machineries, assembled on TFIIIB-bound *tD(GUC)K* tRNA gene promoter in the presence or absence of TFIIIC transcription factor, respectively (Methods and Supplementary Fig. 1d, e).

Using the full integration machinery, we obtained a preliminary negative stain electron-microscopy (EM) reconstruction of the integration complex that showed the presence of an extra density not observed in TFIIIC-TFIIIB/DNA reconstructions, likely corresponding to the Ty3 intasome (Supplementary Fig. 1g). Further analysis by cryo-electron microscopy proved to be unsuccessful, likely due to conformational heterogeneity and low stability of the complex in the conditions used for sample freezing.

Analysis of the minimal integration complex by negative stain EM revealed highly detailed 2D class averages and provided a map where individual domains could be easily discerned and that was successfully used to fit TFIIIB and an in silico model of Ty3 intasome (Supplementary Fig. 1h). We decided to further characterise this complex by cryo-EM. The un-crosslinked sample was applied to carbon-coated cryo-EM grids and data was collected at 0° and 30° tilting angles, to overcome issues with preferred particle orientation (Supplementary Fig. 2a, b). Following a process of hierarchical classification (Supplementary Fig. 2c, d), we obtained a cryo-EM reconstruction of the Ty3

intasome engaged with TFIIIB and the target DNA at an overall resolution of 4.0 Å (Supplementary Fig. 2e-g, Supplementary Fig. 3a and Supplementary Table 1).

The cryo-EM map was used to build an atomic model of Ty3 retrotransposon minimal integration machinery (Fig. 2 and Supplementary Fig. 3b–e). The Ty3 intasome is a pseudo-symmetrical tetrameric complex, in which the four integrase subunits are engaged with two Ty3 long-terminal repeats (LTR) (Fig. 3 and Supplementary Fig. 4a). Two "inner subunits" harbour the two active sites, placing in close juxtaposition the reactive LTR ends with the target DNA and are each associated with an outer subunit (Supplementary Fig. 4a, b). Sequence and structure comparisons show that the Ty3 retrotransposon is more closely related to the Prototype Foamy Virus (PFV)[43] (Supplementary Fig. 4a–c vs. 4d–f) and *S. pombe* Tf1 retroelement than to other retroviruses such as the Rous sarcoma virus (RSV)[44], the β-retrovirus mouse mammary tumour virus (MMTV)[45] or the human immunodeficiency virus (HIV)[46]. Common to these mobile elements is a conserved core consisting of three domains: an HH-CC Zinc binding domain or N-terminal domain (NTD), a catalytic core domain (CCD), which adopts a RNAse H fold, and a SH3 domain or C-terminal domain (CTD) (Fig. 2b and Supplementary Fig. 4c). Additionally, Ty3 encompasses a N-terminal extended domain (NED), which is involved in stabilising the interaction with the LTRs and has been reported to interact with TFIIIC[47] (Supplementary Fig. 4b, c). Intasome binding to the host DNA leads to the formation of a target

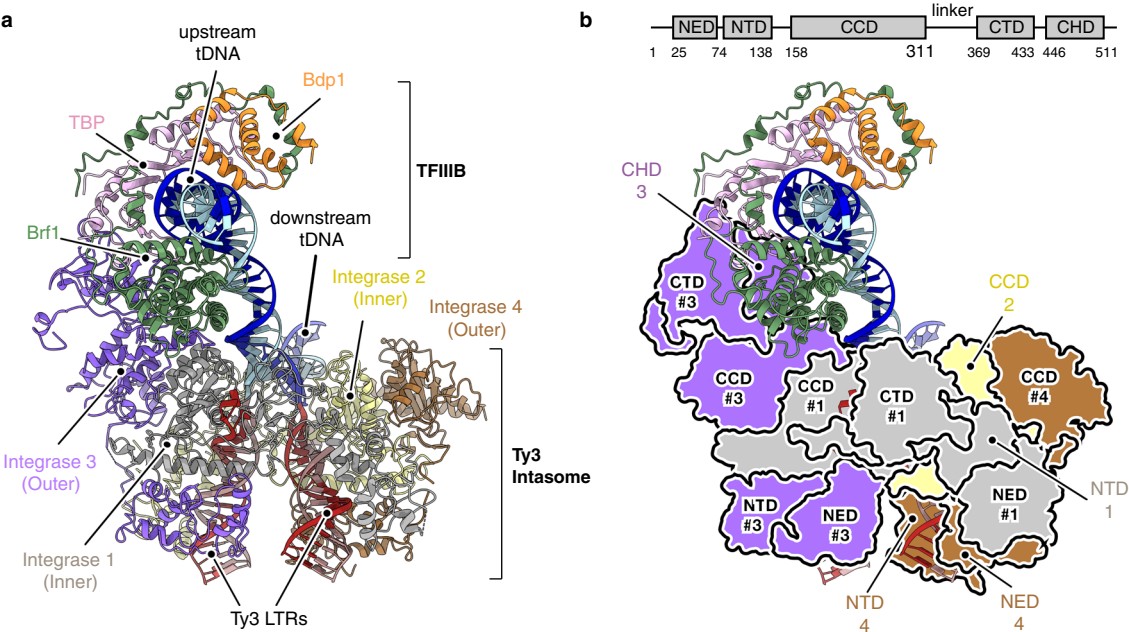

**Fig. 2 Cryo-EM structure of Ty3 strand-transfer complex bound to TFIIIB transcription factor and *tD(GUC)K* tRNA gene. a** Ribbon representation (front view) of Ty3 retrotransposon strand-transfer complex (STC) bound to TFIIIB transcription factor and the *t(GUC)K* gene. Four integrase molecules are numbered and coloured in grey, yellow, purple and brown. Brf1, TBP and Bdp1 subunits of TFIIIB are depicted in green, pink and orange, respectively. Template and non-template strands of the tDNA are shown in dark and light blue, respectively. Ty3 long-terminal repeats (LTRs) are highlighted in red shades. **b** Ty3 retrotransposon model representing as silhouettes the domain architecture of Ty3 integrase subunits (inset). NED N-terminal extended domain, NTD N-terminal domain, CCD catalytic core domain, CTD C-terminal domain, CHD chromodomain.

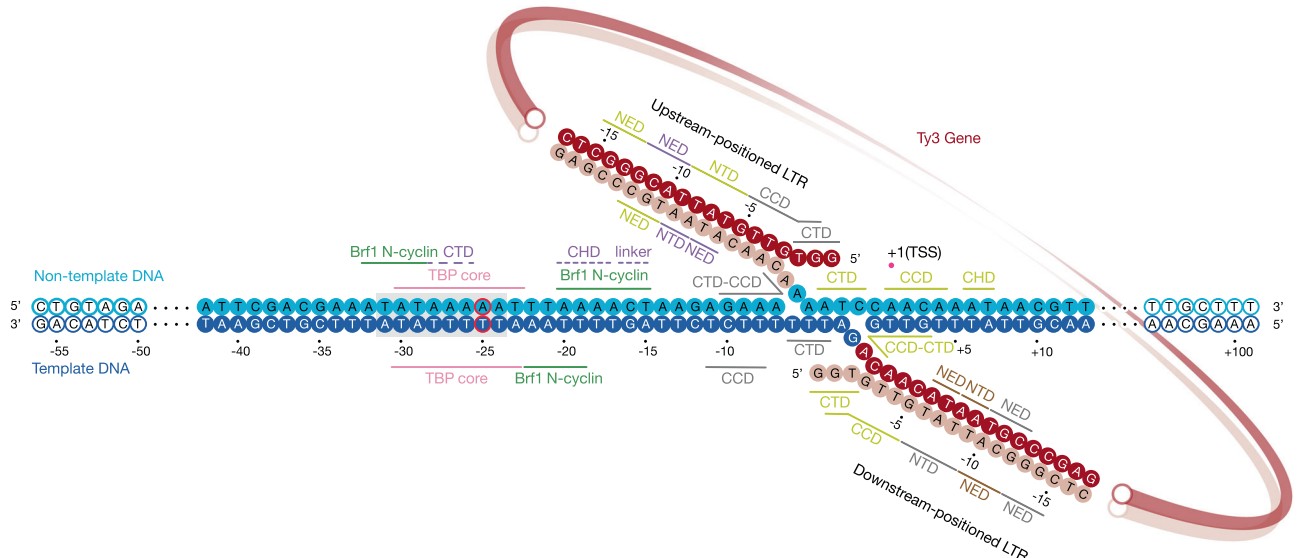

**Fig. 3 Details of protein–DNA interactions and integration site location on RNA Pol III-transcribed genes.** DNA nucleotides of the *tD(GUC)K* gene modelled in the Ty3 strand-transfer complex are depicted as solid circles and numbered relative to the TSS. Template and non-template strands of the tDNA are shown in dark and light blue, respectively. Mutation of nucleotide −25 to break the pseudo-symmetry of the TATA-box is outlined in red. The TATA-box is highlighted with a grey box. Ty3 long-terminal repeats (LTRs) are highlighted in red shades. Protein–DNA and protein–protein interactions are indicated with solid and dashed lines, respectively.

capture complex (TCC) which, after successful catalytic activity and integration of LTRs is referred to as a strand-transfer complex (STC). Analysis of the structure reveals that a phosphodiester bond. has been formed between the 3'-end of the LTR reactive strands and a cleaved 5'-end on the target DNA, confirming the activity of the assembled complex and suggesting that the majority of the particles imaged by cryo-EM correspond to a strand-transfer complex (Supplementary Fig. 3c). The integration reaction takes place at position −6 (relative

to the TSS) of the non-template strand and position −2 (relative to the TSS) of the template strand, which would result in a 5 bp target site duplication (TSD) after a final step of DNA synthesis (Fig. 3). The observed TSD and integration distance from the TSS recapitulate the previous finding observed in vivo[48], supporting the existence of a preferred nucleotide sequence favouring Ty3 integration (Fig. 3).

Ty3 intasome engagement with the target DNA causes a ~90° bent near the transcription start site, which is still compatible

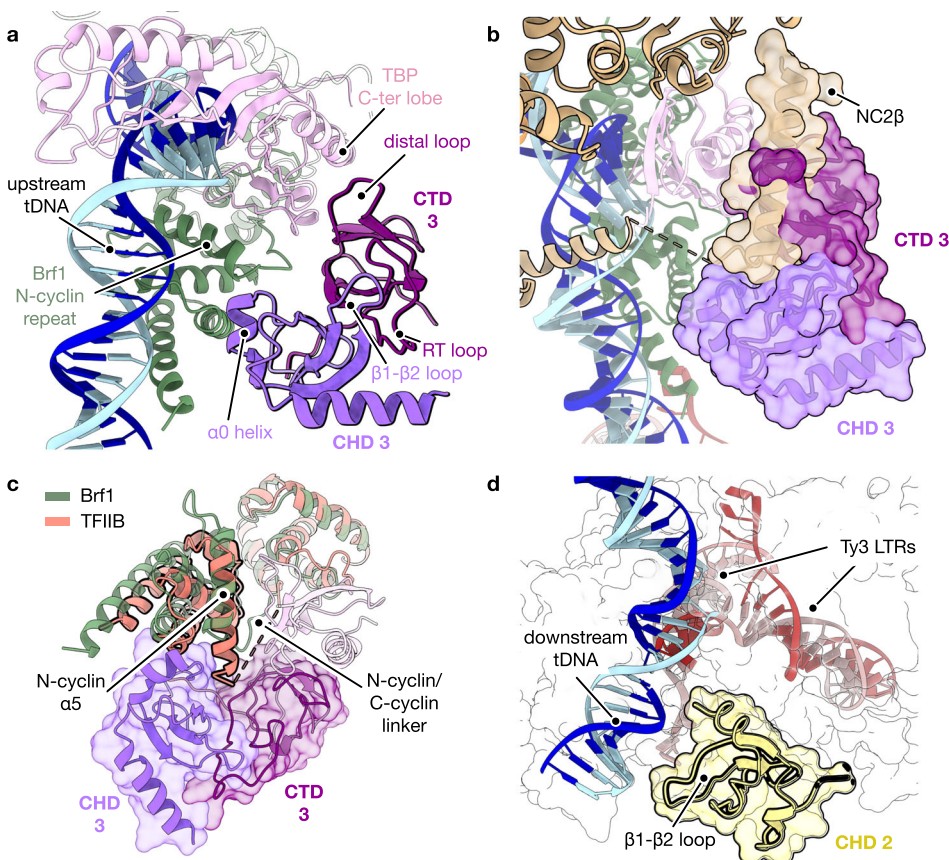

**Fig. 4 Architecture of TFIIIB–Ty3 interaction and molecular basis of integration specificity. a** Detail of TFIIIB transcription factor and Ty3 integrase binding domains, represented as ribbon models. The distal loop of Ty3 integrase C-terminal domain (CTD 3, purple) interacts with TBP C-terminal lobe (pink) whereas Ty3 integrase vestigial chromodomain (CHD 3, violet purple) is recruited to Brf1 N-terminal cyclin repeat (green) through its helix 0 (α0). CTD 3 and CHD 3 domains associate through the RT loop and the β1–β2 loop, respectively, adopting a novel tandem small β-barrel conformation. **b** Binding of the transcription regulator negative cofactor 2 (NC2β, wheat surface, PDB code: 4WZS) and Ty3 intasome to TBP C-terminal lobe requires the same interface, which compromises the recruitment of Ty3 retrotransposon (purple, molecular surface) to RNA Pol II transcriptional units. Structural models are superimposed on TBP molecule (pink ribbon). **c** The B-core cyclin repeat architecture of TFIIB (orange) and Brf1 (green) transcription factors differs in the length of the N-cyclin helix 5 (α5) and the N/C cyclin linker, which prevents binding of Ty3 intasome (purple, molecular surface) to TFIIB and integration at RNA Pol II-transcribed genes. **d** The vestigial chromodomain of a symmetrically related Ty3 integrase subunit (CHD 2, yellow) establishes contacts with the target DNA (tDNA) downstream of Ty3 LTRs integration site.

with binding to the TFIIIB transcription factor, in agreement with recent barcoding experiments highlighting the bendability of this region as an important determinant of frequency of integration[48] (Fig. 2a and Supplementary Fig. 3c). Model building of TFIIIB on the cryo-EM map led to the identification of the TBP core and Brf1 B-core cyclin repeats (Supplementary Fig. 3d, e). However, given the poor resolution around the Bdp1 transcription factor, only a rigid-body fit of the SANT domain could be carried out (Fig. 2a and Supplementary Fig. 3a). Interaction of TFIIIB with the Ty3 intasome is mainly mediated by the C-terminal region of an outer Ty3 integrase subunit, which stablishes a novel set of contacts with the convex surface of TBP stirrups and Brf1 cyclin repeats (Fig. 2). The binding of TFIIIB far upstream (−15 to −32 bp) the TSS constrains the positioning of Ty3 intasome active site also upstream of the TSS, ensuring that RNA Pol III-transcribed genes are not disrupted by the integration event.

**Determinants of target specificity at RNA Pol III-transcribed genes**. Initial model building of the intasome-TFIIIB interface revealed the involvement of the Ty3 integrase C-terminal domain (CTD) (Fig. 4a). The SH3 fold (residues 369–433) of the CTD of an outer Ty3 subunit establishes direct protein–protein

interactions with the H2' helix of TBP C-terminal lobe, positioning the Ty3 intasome upstream the TSS (Fig. 4a). Remarkably, the same region of Ty3 IN, characterised by a compact β-barrel architecture, also mediates tight interactions with the target DNA minor groove when in the context of the inner Ty3 subunits (Fig. 3 and Supplementary Fig. 4b), highlighting the plasticity of the CTD function as previously reported for other retroelements[40]. Topologically, this TBP-Ty3 interface overlaps with the binding interface of TBP and the RNA Pol II-specific transcription regulator negative cofactor 2 (NC2)[49] (Fig. 4b). Thus, NC2β subunit binding to this region of TBP might prevent recruitment of Ty3 intasome to RNA Pol II-transcribed genes before formation of a fully functional pre-initiation complex, rationalising the absence of integration at protein coding genes.

Further inspection of our cryo-EM map revealed the presence of an additional unidentified density close to the outer Ty3 CTD and TFIIIB. De novo model building allowed us to identify this region as a previously uncharacterised chromodomain (CHD), which was thought to be absent in members of the Ty3/Gypsy family in *S. cerevisiae*[50] (Fig. 4a). This domain (residues 446–511) is located C-terminally to the CTD, forming together a novel compact tandem small β-barrel organisation[51], which leads to a tighter binding to the TFIIIB transcription factor (Fig. 4a). Ty3

CHD shares the canonical motif organisation ($\beta_1$-$L_1$-$\beta_2$-$L_2$-$\beta_3$-$L_3$-$\alpha_1$-$\alpha_2$) but it lacks the conserved aromatic cage that is required for histone recognition in similar domains[52] (Supplementary Fig. 5a), likely explaining why it remained unnoticed. Notably, the Ty3 CHD displays an additional N-terminal $\alpha$-helix ($\alpha_0$; aa. 448–456), which folds back into the domain and partially mimics the binding of histone-tail peptides[53–55] (Supplementary Fig. 5b–d). In the context of TFIIIB binding, the $\alpha_0$ helix acts as a "staple" that mediates interactions between the Ty3 integrase and the Brf1 N-terminal cyclin repeat (Fig. 4a). In particular, Ty3 CHD contacts the linker region between Brf1 cyclin repeats, which presents significant differences with TFIIB, the RNA Pol II orthologue of Brf1 (Fig. 4c and Supplementary Fig. 6a). In *S. cerevisiae*, this region of TFIIB includes an unstructured extension which is absent in Brf1 (Supplementary Fig. 6a) and that could represent a hindrance to Ty3 intasome recruitment, disfavouring Ty3 integration at RNA Pol II-transcribed genes.

In our cryo-EM map, additional unattributed density was observed at the downstream end of the target DNA (Fig. 4d). Despite the poorer quality of the map in this region, the density could be unequivocally identified as the CHD of an inner Ty3 integrase subunit. In this context, the Ty3 CHD mediates interactions with the *tD(GUC)K* gene downstream of the transcription start site (Fig. 3 and Fig. 4d). Thus, the positioning of CHDs stemming from two distinct Ty3 subunits at opposite ends of the integration site plays a major role in defining the docking of Ty3 intasome between the TATA-box and the TSS, ensuring that the integration event does not disrupt gene-body regions.

Highlighting the paramount role of CHDs for efficient Ty3 retrotransposition, *YILWTy3-1*, one of two full-length Ty3 genes in *S. cerevisiae*, encompasses a frameshift mutation that disrupts the CHD, resulting in inactivation of this Ty3 retrotransposon that could be re-activated after restoration of the protein reading frame[56].

Analogously to Ty3, CHDs also play a fundamental role in *S. pombe* Tf1 retrotransposon integration, where a non-canonical CHD (Supplementary Fig. 5a) promotes binding to the target DNA and it is required for integration into intergenic regions[8,57,58], pointing towards a more general role of CHDs for efficient transposition of Ty3/Gypsy retroelements across different species.

**Ty3 intasome recruitment by TFIIIB requires an extended CCD-CTD linker.** Recruitment of Ty3 intasome to RNA Pol III-transcribed genes depends on the interaction between TFIIIB and Ty3 integrase C-terminal region, which questions whether other CTD-containing retroelements could exploit similar mechanisms to target transcription factors and/or DNA bound complexes. Among the conserved integrase core domains, one of the main differences between Ty3 and other integrases such as RSV, MMTV or HIV, lays on the presence of a longer CCD-CTD linker (Supplementary Fig. 6b). This linker folds into an elongated $\alpha$-helix ($\alpha_{10}$) and this feature is especially significant in the context of TFIIIB targeting because it allows the positioning of Ty3 CTD and CHD at the outer boundaries of the intasome, acting as a platform for TFIIIB anchoring (Fig. 5a). Due to the reduced length of the CTD-CCD linker in RSV, HIV and MMTV

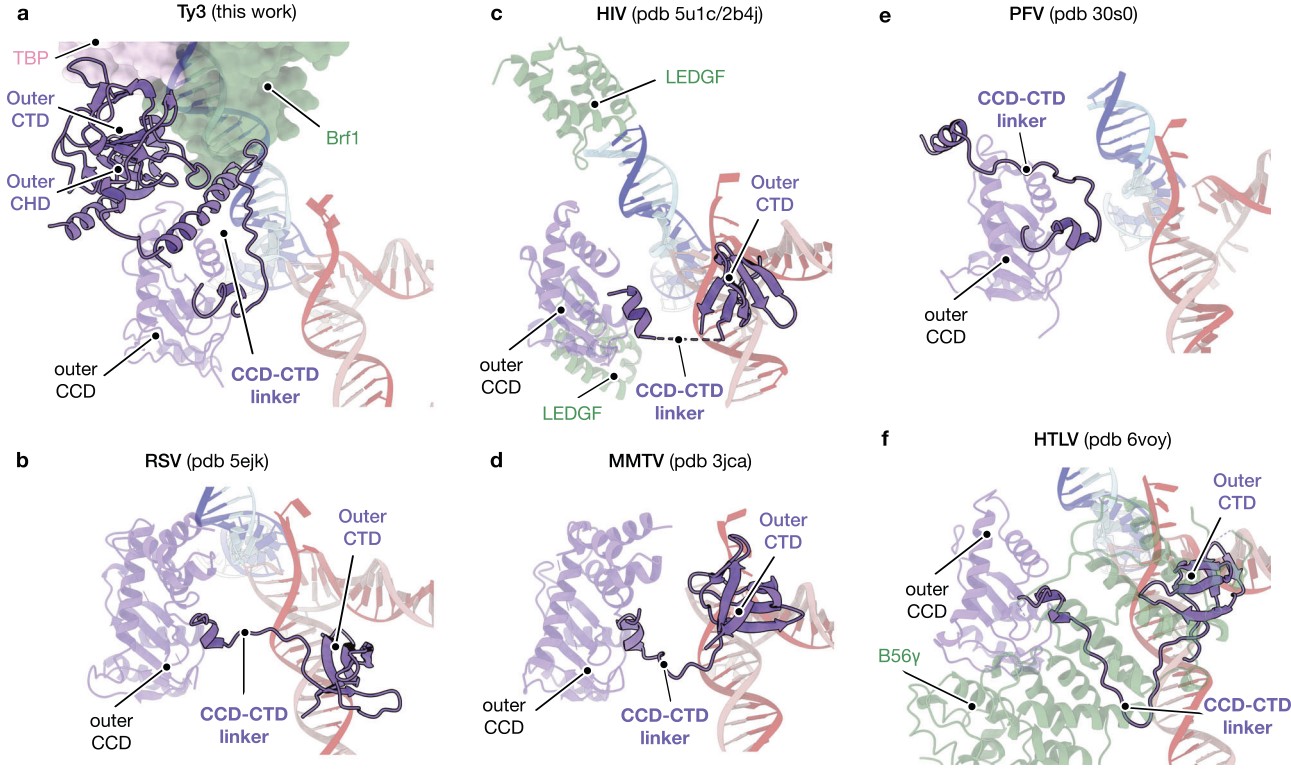

**Fig. 5 Ty3 CCD-CTD extended linker creates a platform for TFIIIB assembly. a** Detailed view of the orientation adopted by the CCD-CTD linker of Ty3 integrase outer subunit. Ty3 extended linker adopts a peripheral orientation that creates a platform for the interaction between Ty3 CTD and CHD domains (purple ribbon) and TFIIIB transcription factor (pink and green surfaces). The outer catalytic core domain (CCD) and the DNA molecules are shown as transparent ribbons. **b** As in **a** but for Respiratory Syncytial Virus (RSV, PDB code: 5EJK). **c** As in **a** but for Human Immunodeficiency Virus (HIV, PDB code: 5U1C). Binding of LEDGF/p75 transcription factor (green ribbon) to HIV could be mediated by a similar region than in the Brf1-IN interaction, which highlights a potential hot-spot for host-integrase recognition events. **d** As in **a** but for Mouse Mammary Tumor Virus (MMTV, PDB code: 3JCA). **e** As in **a** but for Prototype Foamy Virus (PFV, PDB code: 3OS0). **f** As in **a** but for Human T-lymphotropic Virus Type 1 (HLTV, PDB code: 6VOY). Binding of CCD-CTD linker to B56$\gamma$ (green ribbon) is represented.

retroviruses, their CTDs are positioned close to the intasome core, which prevents the interaction with factors positioned away from the body of the intasome[44–46] (Fig. 5b–d). Conversely, the PFV retrovirus CTD-CCD linker displays a similar length and adopts a similar orientation compared to Ty3 integrase, suggesting targeting of more distant factors[59] (Fig. 5e and Supplementary Fig. 6b). Although the presence of alpha-helical CCD-CTD linkers is observed in lentiviral integrases such as HIV-1, SIV or MVV[60,61], the use of this folded element as a platform for recruitment to host factors has not been described before. Interestingly, in spite of the lack of alpha-helical organisation, the CCD-CTD linker of HTLV-I integrase participates in the recruitment to β56 factor (Fig. 5f).

The overall architecture of the Ty3 intasome resembles the organisation of other retroelements such as PFV (Supplementary Fig. 4). Intriguingly, despite the differences in the outer integrase subunits, the binding mode of Ty3 intasome to TFIIIB transcription factor resembles the one of the tetrameric HIV intasome to one of the potential binding sites for the lens epithelium-derived growth factor (LEDGF/p75) (Fig. 5c). Similar to Brf1 B-core cyclin repeats, the integrase binding domain of LEDGF/p75 factor also adopts a helical bundle organisation (PDB codes: 5U1C and 2B4J)[16,46]. This factor is required for HIV targeting host genomes and its depletion prevents retroviral integration and replication[17,62–64]. Despite the distant evolutionary link between retroelements, the use of similar interfaces hints at the existence of "hot-spot" regions at the intasome periphery which are exploited to target DNA-binding factors in the host organism.

**Ty3 integration and RNA Pol III transcription are mutually exclusive.** The central role of TFIIIB factor in Pol III recruitment and transcription initiation has been extensively characterised[21–25,65] but it remains to be understood how this is affected by other TFIIIB-mediated processes occurring at the same loci, such as Ty3 retrotransposon integration. The cryo-EM structure of TFIIIB targeted by a Ty3 intasome allowed us to compare these two unrelated processes. Superimposition of Ty3-TFIIIB complex with that of the Pol III pre-initiation complex (PIC)[21] (Fig. 6) reveals that the architecture of TFIIIB is unaltered, except for the destabilisation of Bdp1 binding in the retrotransposon targeting complex reconstituted in vitro, which is essential for DNA opening during transcription initiation. Furthermore, the comparison of the two structures reveals that the Ty3 intasome and Pol III would occupy nearly the exact same position, thus providing mechanistic evidence that Pol III transcription and Ty3 integration are mutually exclusive, as previously postulated by studies suggesting competition between the two processes[48,66]. Unlike Ty3, Ty1 retrotransposon integrates at nucleosomal DNA and requires a physical association with the Pol III enzyme[13,14,67] rationalising why Ty3 and Ty1 do not compete for the same integration sites.

**Discussion**
Retrotransposons are the main active TEs in eukaryotes and are structurally and functionally related to retroviruses. The selection of integration sites clearly impacts on the consequences of mobilisation of TE elements on the host biology. Thus, insertions in deleterious locations are negatively selected. In this context, the Ty3 family of TE has evolved exploiting the peculiar architecture of Pol III-transcribed genes, which are defined by ubiquitous internal promoter elements such as the A and B boxes, which are binding sites of transcription factor TFIIIC[68–70]. By tightly binding to upstream transcription factor TFIIIB, which is ultimately recruited by TFIIIC, the Ty3 intasome drives the insertion

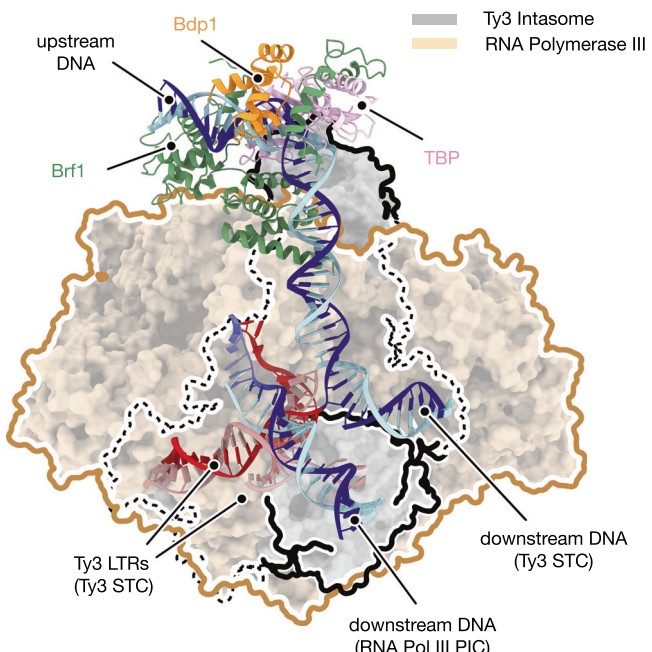

**Fig. 6 General comparison of TFIIIB targeting by RNA Polymerase III and Ty3 intasome.** Comparison of RNA Polymerase III and Ty3 intasome recruitment by TFIIIB transcription factor. TFIIIB subunits are depicted as ribbons and coloured as described in Fig. 2. *S. cerevisiae* models of RNA Pol III pre-initiation complex (RNA Pol III PIC, PDB code: 6EU0) and Ty3 strand-transfer complex bound to TFIIIB (Ty3 STC, this work) were superimposed onto TBP subunit of TFIIIB. RNA Pol III and Ty3 intasome are depicted as wheat and grey molecular surfaces, and their boundaries are highlighted with brown and black lines, respectively. Binding of both enzymatic machineries to RNA Pol III-transcribed genes requires equivalent regions, which prevents simultaneous RNA transcription and integration events. Models of the target DNA are shown as ribbon and coloured as described in Fig. 2. Whereas the orientation of the upstream DNA is conserved, the direction of the tDNA in the proximity of the TSS differs upon binding of the two enzymes.

of new copies of the Ty3 gene upstream of the transcribed region, while concomitantly leaving the promoter elements untouched. This strategy confers Ty3 retrotransposon the ability to integrate into safe-haven locations of the host genome while minimising deleterious effects.

Our cryo-EM reconstruction of Ty3 intasome bound to a tRNA gene promoter recapitulates previous biochemical data and integrates them into a model that involves the formation of an extensive network of interactions between TFIIIB transcription factor and an outer Ty3 integrase subunit. The structural data unveils how a newly discovered chromodomain in Ty3 integrase mediates histone-independent interactions with Brf1 and the downstream part of the target DNA, acting as a "ruler" that safeguards the integration upstream of the transcription start site and prevents gene disruption (Fig. 3 and Fig. 4). Our atomic model deciphers the molecular determinants of target specificity at RNA Pol III-transcribed genes, highlighting how Ty3 intasome recruitment requires a region in the Brf1-TBP interface that is blocked by the NC2β factor in the TFIIB homologous factor at RNA Pol II-transcribed genes (Fig. 4). The incompatibility between Pol III machinery activity and Ty3 integration additionally allows for these two processes to be timely separated avoiding a direct overlap that could result in dangerous loss of genetic information.

Notably, our results found that Brf1 targeting by Ty3 element shows similarities to the recruitment of HIV to LEDGF/p75

factor, suggesting that host recognition occurs through similar mechanisms and interfaces across distant retroelements.

## Methods

**Protein expression and purification.** A pOPINF plasmid containing *S. cerevisiae* full-length Ty3 integrase was transformed into *E. coli* BL21-RIL-CodonPlus competent cells and grown at 37 °C to OD$_{600}$ of ~1 in LB medium supplemented with 50 μM ZnCl$_2$. The cell culture was stored at 4 °C for 1 h and an overnight induction was performed with 1 mM IPTG at 20 °C. Following cell harvest, the collected pellet was suspended in lysis buffer containing 20 mM HEPES pH 8, 750 mM NaCl, 10 mM MgCl$_2$, 10 μM ZnCl$_2$, 10 mM β-mercaptoethanol, 10 mM imidazole, 50 mM L-arginine and 50 mM L-glutamic acid, supplemented with two protease inhibitor tablets (Roche) and a scoop of DNAse, and adjusted to pH 7.4. After incubation for 30 min at 4 °C, cells were subjected to sonication (12 cycles, 15 s ON, 59 s OFF, 60% amplitude) and the lysate was fractionated by centrifugation at 48,000 × *g* for 40 min at 4 °C. Then, the soluble fraction was filtered and incubated for 3 h at 4 °C with ~6 ml of HisPur™ Ni-NTA resin (Thermo Fisher Scientific). The beads were subsequently washed with ~400 ml lysis buffer (without protease inhibitors or Arginine/Glutamic acid) and the protein eluted in 20 ml lysis buffer supplemented with 500 mM imidazole. N-terminal His-tag was removed by overnight incubation with ~1 mg of 3 C protease at 4 °C. The cleaved sample was then diluted to 125 mM NaCl with buffer HepA (20 mM HEPES pH 8, 10 mM MgCl$_2$, 10 μM ZnCl$_2$ and 1 mM DTT) and loaded into a HiTrap Heparin HP 5 ml column (GE Healthcare) equilibrated with 6.25% of buffer HepB (buffer HepA supplemented with 2 M NaCl). After washing, sample elution was performed through a linear gradient from 6.25% to 100% of buffer HepB in 30 CV. Fractions corresponding to the elution peak were analysed by SDS-PAGE and those containing Ty3 integrase were pooled, concentrated to ~4 ml and loaded into a HiLoad 16/600 Superdex 200 pg column (GE Heathcare) equilibrated with 20 mM HEPES pH 8, 730 mM NaCl, 10 mM MgCl$_2$, 10 μM ZnCl$_2$ and 1 mM DTT. The fractions corresponding to pure Ty3 integrase were pooled, concentrated to ~3.25 mg/ml, flash-frozen in liquid nitrogen and stored at −80 °C. The final yield was ~2 mg of protein per litre of culture (Supplementary Fig. 1a).

The six-subunit TFIIIC transcription factor was cloned into pBIG2ab vector using the biGBac system[71]. After plasmid transformation into the DH10EmBacY strain using the heat-shock technique, the cells were plated on an agar plate containing X-Gal, gentamycin, kanamycin and tetracycline and incubated overnight at 37 °C. White colonies corresponding to successful plasmid transposition were selected, expanded in 5 ml of LB media and the bacmid containing the TFIIIC genes was purified. The generated plasmid was then mixed with Cellfectin™ II reagent (Thermo Fisher Scientific) in a 1:1 molar ratio and transfected into adherent Sf9 insect cells at a density of 5 × 10$^5$ cells per millilitre. Following incubation at 27 °C for 72 h, the cell culture supernatant (~2 ml), which contained the P1 viral fraction, was collected and mixed with 25 ml of suspended Sf9 cells at a density of 5 × 10$^5$ cells per millilitre. Cell viability and visualisation of the yellow fluorescent protein (YFP) in the cells for 3–5 days were used as an indicator of viral infection. The supernatant corresponding to the P2 viral fraction was collected once fluorescence was observed in 80–90% of the cells. Finally, protein expression was achieved by the addition of 2 ml of P2 fraction to 500 ml of suspended High5 insect cells at a density of 5 × 10$^5$ cells/ml and growth for 3–4 days at 27 °C in Lonza Insect Xpress media. Cells were harvested and the pellet stored at −80 °C.

The insect cells pellet was suspended in ~125 ml of lysis buffer containing 20 mM HEPES pH 8, 500 mM NaCl, 1 mM MgCl$_2$, 10% glycerol and 10 mM β-mercaptoethanol, supplemented with a scoop of DNase I, two protease inhibitor tablets (Thermo Fisher Scientific) and 4 μl of benzonase. After incubation, the sample was subjected to sonication (9 cycles, 5 s ON, 10 s OFF, 20% amplitude) and fractionated by centrifugation at 37,565 × *g* for 30 min at 4 °C. The soluble fraction was filtered and loaded into a StrepTrap HP 5 ml column (GE Healthcare) equilibrated with lysis buffer. Following a wash step, the bound sample was eluted in 15 ml of lysis buffer supplemented with 0.05% (w/v) D-desthiobiotin. The sample was diluted to ~150 mM NaCl with buffer HepA (20 mM HEPES pH 8, 10% glycerol and 10 mM β-mercaptoethanol) and loaded into a HiTrap Heparin HP 5 ml column equilibrated with 7.5% of buffer HepB (buffer HepA supplemented with 2 M NaCl). After washing, elution from the column was performed through a linear gradient from 7.5% to 100% of buffer HepB in 10 CV. The fractions corresponding to the elution peak were analysed by SDS-PAGE and those containing the TFIIIC complex were collected and loaded into a XK 16/70 Superose 6 pg column (GE Healthcare) equilibrated with 20 mM HEPES pH 8, 200 mM NaCl, 2.5% glycerol and 1 mM DTT. Pure TFIIIC fractions were pooled, concentrated to ~12.5 mg/ml, flash-frozen and stored at −80 °C (Supplementary Fig. 1a).

Purifications of *S. cerevisiae* Brf1-TBP fusion protein and Bdp1 transcription factor were performed as described elsewhere[21] (Supplementary Fig. 1a).

***tD(GUC)K* target DNA.** Fluorescently-labelled DNA fragments used for in vitro binding and integration assays were generated by large-scale PCR amplification of the *tD(GUC)K* promoter between -56 and +101 (according to TSS position) using *Pyrococcus abysii* DNA polymerase (PabPolB) purified in house. The generated

DNA was precipitated with ice cold 100% ethanol (v/v), suspended in buffer QA (10 mM Tris pH 8 and 1 mM EDTA) and loaded into a MonoQ 5/50 GL column (GE Healthcare). After a step-wise wash with 10% and 20% of buffer QB (Buffer QA supplemented with 2 M NaCl), elution was performed through a linear gradient from 20 to 50% of buffer QB in 40 CV. After analysis in 2% agarose gels, fractions containing pure DNA were pooled, precipitated with ice cold 100% ethanol and suspended in TE buffer pH 8 to a final concentration of ~70 μM. The resulting 157-bp DNA fragment included the TATA-box, and the A- and B-box elements required for TFIIIC binding. Labelling of the upstream or downstream termini with Cy3 or FITC, respectively, was achieved through incorporation of the fluorophores in the amplification oligonucleotides.

Target DNA oligonucleotides used for cryo-EM analysis of Ty3 intasome binding to TFIIIB transcription factor included the TATA-box and encompassed nucleotides −42 to +13 (according to TSS position) of the yeast *tD(GUC)K* gene promoter (template strand: 5'- AACGTTATTTGTTGGATTTTTTCTCTTAGTT TTAAATTTTTATATTTCGTCGAAT -3' and non-template strand: 5'- ATTCGA CGAAATATAAAAATTTAAAACTAAGAGAAAAAATCCAACAAATAACGT T -3'; Integrated DNA Technologies). As described previously[72], a single nucleotide mutation in the TATA-box element was included in order to favour unidirectional positioning of TFIIIB.

**Ty3 intasome complex formation.** The Ty3 intasome assembly was carried out using a 5'-overhang fragment of the retrotransposon LTR terminus, which consisted of a 17 bp reactive strand (5'-GAGCCCGTAATACAACA-3'; IDT) and a 19 bp non-reactive strand (5'-GGTGTTGTATT ACGGGCTC-3'; IDT). For in vitro binding and activity assays, a Cy5 fluorophore at the 5' end of the reactive strand was included. The oligonucleotides were suspended and annealed as previously described[21]. The intasome assembly was carried out using 1.25 nmol of annealed LTRs and a 2.4-fold excess of pure Ty3 integrase, in 425 μl of buffer containing 20 mM HEPES pH 8, 730 mM NaCl, 10 mM MgCl$_2$, 10 μM ZnCl$_2$ and 1 mM DTT. The mixture was subjected to a progressive reduction of the salt concentration through overnight dialysis in 20 mM HEPES pH 8, 200 mM NaCl, 25 μM ZnCl$_2$ and 2 mM DTT, using a 1 kDa MWCO Mini Dialysis Kit (GE Healthcare). Finally, the concentration of NaCl in the sample was increased to 320 mM by addition of NaCl 5 M. Intasome formation was assessed by gel-filtration chromatography (Supplementary Fig. 1b) followed by SDS-PAGE analysis and detection of the Cy5 fluorophore in a native agarose gel (Supplementary Fig. 1c).

**Assembly of full and minimal integration machinery.** In vitro targeting of the Ty3 intasome at the full RNA Pol III transcriptional machinery (i.e. containing TFIIIC and TFIIIB) was achieved through a process of sequential binding of transcription factors and integration machinery to the target DNA. First, 200 μg of TFIIIC transcription factor and the annealed 157 bp *tD(GUC)K* DNA were mixed in a 2.2:1 molar ratio and incubated at RT for 30 min (Fig. 1b, lane 2). Next, a 2.5-fold excess (relative to the DNA) of Brf1-TBP fusion protein and Bdp1 were added to the mixture in two consecutive steps and subjected to a similar incubation protocol (Fig. 1b, lanes 3–4). Finally, the reaction was supplemented with 400 μl of Cy5-labelled Ty3 intasome (assembled as described before) and the targeting allowed for 30 min at RT (Fig. 1b, lane 5). Controls were performed with unlabelled Ty3 intasome (Fig. 1b, lane 6), with Ty3 intasome buffer (Fig. 1b, lane 7) or in the absence of transcription factors (Fig. 1b, lanes 8–9). After each binding step, samples were collected and analysed by electrophoretic mobility shift assays (EMSAs) in 1% agarose gels. Detection of the Cy5 and FITC fluorophores present in the LTR and target DNA, respectively, was performed using a Typhoon FLA 9000 gel imaging scanner (GE Healthcare). For cryo-EM analysis, the sample was concentrated to ~400 μl, centrifuged and loaded into a Superose 6 increase 10/300 GL column (GE Healthcare) equilibrated with 40 mM Tris pH 7, 80 mM NaCl, 7 mM MgCl$_2$ and 1 mM DTT (Supplementary Fig. 1d). Fractions corresponding to the first elution peak were pooled.

Assembly of the minimal integration machinery was performed in the absence of the TFIIIC transcription factor following the same protocol but using an unlabelled 55-bp target DNA (Supplementary Fig. 1e-f). Fractions corresponding to the first elution peak were pooled, concentrated to ~0.01 mg/ml with an Amicon Ultra 0.5 30 K NMWL (Thermo Fisher Sci.) centrifugal concentrator and used for subsequent EM analysis. All uncropped gel images are available in the Source Data file.

**Ty3 integration assay.** Evaluation of the integration activity of the reconstituted Ty3 intasome was performed through analysis of the DNA products after targeting at the full RNA Polymerase III transcriptional machinery. Samples were collected after recruitment of TFIIIC, TFIIIB and Ty3 intasome to the *t(GUC)K* target DNA, which was Cy3- or FITC-labelled in the upstream or downstream terminus, respectively (Fig. 1a). Reaction mixtures were prepared in the presence of Cy5-labelled or unlabelled intasomes and, as a control, the assay was also performed in the absence of integration machinery (Fig. 1c, d). In order to analyse the nucleic acid fraction, the protein content was degraded using the broad-spectrum serine protease proteinase K. Thus, the sample was mixed with a proteinase K mixture (2 mg/ml proteinase K, 2% SDS (w/v) and 200 mM EDTA pH 8) in a 10:1.1 volume ratio and incubated for 1.5 h at 37 °C. Then, the cleaved sample was

loaded into a 4% agarose gel and identification of the integration products was performed through detection of the FITC, Cy5 and Cy3 fluorophores in a Typhoon FLA 9000 gel imaging scanner (GE Healthcare) (Fig. 1c, d). 50 bp DNA markers (N3236S, New England Biolabs) were used to identify the size of the resulting DNA products (Fig. 1e). All uncropped gel images are available in the Source Data file.

**Negative stain EM sample preparation, data collection and processing**. Negative stain electron microscopy of the full integration complex (TFIIIB-TFIIIC-DNA-Intasome; 920 kDa; Supplementary Fig. 1g), the minimal integration complex (TFIIIB-Intasome; 400 kDa; Supplementary Fig. 1h) and the TFIIIB-TFIIIC-DNA complex (670 kDa; Supplementary Fig. 1g) were performed after gel-filtration chromatography following the same protocol. In brief, 3 μl of sample were applied to glow discharged (1 min at 15 mA; PELCO EasiGlow) Quantifoil R 1.2/1.3 copper grids coated with a thin carbon film, incubated for 1 min, washed with ultra-filtered water and stained with 2% uranyl acetate for 30 sec. Data collection was performed in-house in a ThermoFisher Tecnai F20 TEM microscope operating at 200 kV, equipped with a F416 CMOS camera (TVIPS) and at a ×50,000 magnification corresponding to 1.732 Å per pixel. All data processing was carried out in Relion 3.0.4.

**Cryo-EM sample preparation and data collection**. Cryo-EM samples of *S. cerevisiae* Ty3 intasome engaged with TFIIIB and the *tD(GUC)K* target DNA were prepared on Quantifoil R 1.2/1.3 (400 mesh) copper grids coated with a thin carbon film prepared in house. Grids were glow discharged for 30 s at 15 mA (PELCO EasyGlow) before the application of 2 μl of sample at ~0.1 mg/ml. After 30 s incubation at 18 °C and 100% humidity, the grids were blotted (drain time: 0.5 s, blot force: 3, blot time: 3 s) and plunged frozen into liquid ethane using a Vitrobot Mark IV system (ThermoFisher).

Separate data collections were performed for un-tilted and 30°-tilted datasets at the Electron Bio-Imaging Centre (eBIC) on a Titan Krios transmission electron microscope (ThermoFisher) at 300 KeV using EPU2 Software. For the un-tilted data collection, 4974 movies were collected using a K2 Summit direct electron detector (Gatan, Inc.) at a 1.47 Å calibrated pixel size and a nominal magnification of ×130,000 (Supplementary Fig. 2a). Fifty frames were collected per movie at a defocus range between −1.7 and −3.2 μm and a total exposure of 7 s (dose rate of 7.14 e−/ Å$^2$/s, an accumulated total dose of ~50 e−/ Å$^2$ and a fractionated dose per frame of 1 e−/ Å$^2$). For the 30°-tilted data collection, 2437 movies were collected in a K3 direct electron detector (Gatan, Inc.) at a nominal magnification of ×81,000 and a calibrated pixel size of 0.53 Å (Supplementary Fig. 2b). Each movie was fractionated over 70 frames and collected for 5.3 s at a dose per frame of 1 e−/ Å$^2$ and at a defocus range from −2.4 to −4.0 μm, which yielded a dose rate of 13.20 e−/ Å$^2$/s and an accumulated total dose of ~70 e−/ Å$^2$.

**Cryo-EM image processing**. Frame alignment and dose-weighting steps were performed on-the-fly during data collection using MotionCor2 software[73] and CFFIND4[74] was used for the estimation of the contrast transfer function (CTF) parameters. Unless stated otherwise, all the subsequent steps of particle picking, extraction, 2D and 3D classification, and post-processing were carried out in Relion (versions 3.0.4–3.1.1)[75].

For the un-tilted dataset, 1,090,240 particles were auto-picked and subjected to three steps of classification that provided highly detailed 2D class averages, which accounted for a subset of 386,471 particles. Then, the particles were imported into CryoSPARC 2.0[76] and subjected to an ab-initio 3D classification process, which provided five major classes. Class #0 (~86,939 particles) and class #4 (~94,355 particles) showed the characteristic tetrameric organisation observed in other intasome reconstructions (i.e. PFV or HIV) and the presence of an extra unidentified density. After joining the particles from these two classes, a hetero-refinement step was carried out, which resulted in two new 3D reconstructions: class #0 (~55,546 particles) at 9.65 Å-resolution and class #1 (~125,748 particles) at 5.40 Å-resolution. Particles corresponding to class #1 were re-extracted and imported into Relion, where they were subjected to a refinement and post-processing step that resulted in a final 3.8 Å-resolution map, according to the gold-standard FSC cut-off criterion at 0.143. The resulting map showed unequivocally the presence of the Ty3 intasome complex (containing four molecules of Ty3 integrase and two LTR fragments) engaged with the target DNA fragment and the Brf1 and TBP transcription factors. However, detailed analysis of the map and of the particle orientation distribution sphere indicated the existence of preferential orientation, which caused the presence of anisotropic features in the reconstruction.

For the 30°-tilted dataset, 994,999 particles were auto-picked, extracted and subjected to five steps of 2D classification, which resulted in a subset of 242,527 particles. 3D classification of this subset provided three classes that showed anisotropic characteristics similar to those observed previously in the un-tilted dataset.

With the aim of solving these problems, the un-tilted and 30°-tilted datasets were merged following a protocol described elsewhere[77]. In brief, after calculating the "real" pixel size from the obtained 3D reconstructions, particles from the un-tilted 2D-classification subset were re-extracted and re-sized to the nominal pixel size of 1.06 Å observed in the tilted dataset. Then, the scaled particles were merged

with the 2D-classification subset of the tilted data collection, resulting in a new dataset of 628,998 particles (Supplementary Fig. 2c). Next, 3D classification was performed using a 20 Å-filtered map obtained from the un-tilted dataset, which resulted in five different classes (Supplementary Fig. 2d). Class #3 (169,140 particles), Class #4 (116,343 particles) and Class #5 (118,081 particles) showed density features corresponding to Ty3 intasome and TFIIIB. After joining the particles, a 3D refinement was performed, which rendered a map at 3.97 Å-resolution, according to the gold-standard FSC cut-off criterion at 0.143. Then, to improve the quality around specific regions of the map, we performed a hierarchical 3D classification process without alignment using masks around the intasome, TFIIIB and the outer CTD-CHD domains (Supplementary Fig. 2d). The resulting class was subjected to a CTF refinement process that encompassed trefoil and 4th order aberration corrections, followed by correction of magnification anisotropy and finally, defocus refinement on a per particle basis to correct CTF estimation errors for the titled particles. Next, a symmetry expansion process followed by further consensus refinement and post-processing steps rendered a map which showed improved features and density corresponding to side-chains. Local resolution analysis reported a global resolution of 3.98 Å-resolution at the gold-standard 0.143 FSC cut-off criterion (Supplementary Fig. 2e-g).

**Cryo-EM model building and refinement**. An initial model of Ty3 integrase was obtained using the Phyre2 modelling web portal[78], which reported structural similarity to the prototype foamy virus (PFV) integrase. Four copies of the predicted Ty3 integrase were structurally aligned to the tetrameric structure of PFV[59] (RCSB Protein Data Bank (PDB) code: 3OS0), which rendered a preliminary in silico model of the Ty3 intasome. A rigid-body fit of the model into the cryo-EM map was subsequently performed in UCSF Chimera (version 1.14)[79], which provided the estimated position and general orientation of Ty3 intasome. In order to account for differences in the relative orientation of the Ty3 integrase domains, each subunit was split into its constituting domains: N-terminal extended domain (NED), N-terminal domain (NTD), catalytic core domain (CCD) and C-terminal domains (CTD). The resulting individual domains were rigidly fitted into the cryo-EM map in Chimera, which led to an improved positioning of the subunits. Manual correction of the connecting regions was carried out in Coot (version 0.8.9.2)[80] and it was guided by secondary structure prediction performed in the PsiPred webserver (http://bioinf.cs.ucl.ac.uk/psipred/). Following manual fitting, the model was subjected to real-space refinement in the Phenix Suite (version 1.18.1–3865)[81] and the process was repeated iteratively until a tetrameric model of Ty3 intasome was obtained.

At this stage, regions of unassigned density were still observed in the map and we proceeded to the rigid-body fit of the TFIIIB transcription factor (PDB code: 6EU0) into them using Chimera. We carried out a manual and automatic refinement in Coot and Phenix, as previously described, which allowed us to build the majority of the Brf1 and TBP models. Density corresponding to Bdp1 protein was very poor and only a rigid-body fit of the SANT domain was performed using Chimera.

Next, DNA fragments of the PFV gene LTRs (PDB code: 3OS0) were used as a reference model and rigidly fitted into the map using Chimera. Following in silico mutagenesis to correct the sequence discrepancies, the Ty3 LTRs were real-space refined in Phenix. Similarly, model building of the target DNA was based on structural alignment of TFIIIB bound to the *SNR6* promoter (PDB code: 6EU0), which allowed us to determine the position of the TATA-box element. Nucleotides were then mutated into the *tD(GUC)K* promoter sequence. Next, the target DNA model was extended by an iterative process consisting of the manual addition of 3–4 nucleotides at the downstream end and its refinement in Phenix. This method was repeated recursively until the DNA molecule near the intasome active site was built. Here, the structure of HIV strand-transfer complex[46] (PDB code: 5U1C) was used as a reference to determine the position of the LTR reactive strands and to build the 5-nt target site duplication (TSD), which is characteristic of Ty3 intasome. Previous studies into the local features of Ty3 targeting[48] were taken into account in order to ensure the nucleotide registry around the integration sites was correctly modelled.

After model building of TFIIIB, Ty3 intasome and the target DNA, we observed the presence of two extra densities in the cryo-EM map: one close to the TFIIIB transcription factor and another near the DNA downstream end. After building of the Ty3 intasome, the C-terminal domain (CTD) of an outer integrase subunit remained unmodelled, since its position differed from that observed in similar structures. However, detailed analysis of the unassigned densities showed that the missing Ty3 CTD could be rigidly-fit with high confidence into the extra map near TFIIIB. Following docking, the linker to the catalytic core domain (CCD) was built using Coot and this region was subjected to real-space refinement in Phenix. After this process, part of the extra map near Brf1 still remained unidentified. This density was connected to the newly built CTD and it seemed to correspond to the very C-terminus of the Ty3 integrase, a region absent in retroviral homologs. In order to determine the protein architecture of this region (aa. 440–504), an intensive in silico modelling was carried out in Phyre2 which predicted its folding into a chromodomain (CHD). The generated Ty3 CHD was accurately fitted into the map near Brf1 using Chimera, and refined in Phenix after manual correction in Coot.

Finally, the remaining unassigned density near the downstream DNA region was also situated close to the C-terminal domain of an inner integrase subunit, which prompted us to consider whether this feature also corresponded to a chromodomain. Albeit the map quality around this area was poor, rigid-body fit of the CHD model into the unsharpened cryo-EM map unequivocally identified this density as a chromodomain fold.

Cryo-EM Figures were prepared using UCSF Chimera 1.14 and ChimeraX-1.0. Multiple sequence alignments were calculated using the Clustalw Omega server (https://www.ebi.ac.uk/Tools/msa/clustalo/) and Figures were prepared in Jalview 2.10.1. Structural comparisons were performed in Pymol v1.8.6.0 from the corresponding PDB codes obtained in the RCSB database (https://www.rcsb.org/).

**Reporting summary**. Further information on research design is available in the Nature Research Reporting Summary linked to this article.

## Data availability

The data that support this study are available from the corresponding authors upon reasonable request. The structure of the Ty3 intasome engaged with TFIIIB and the tRNA promoter and its associated data have been deposited into the Protein Data Bank under accession code 7Q5B, and into the Electron Microscopy Data Bank under accession code EMD-13831. Additional atomic models used in this study correspond to: PFV strand-transfer complex (3OS0), HIV-1 Strand Transfer Complex (5U1C), HIV-1 integrase and LEDGF/p75 (2B4J), RSV intasome (5EJK), Mouse Mammary Tumour Virus intasome (3JCA), HTLV-1 intasome (6VOY), RNA Polymerase III open pre-initiation complex (6EU0), NC2 in complex with TBP and Mot1 N-terminal domain (4WZS), Polycomb chromodomain complexed with H3K27me3 histone tail (1PDQ), HP1a chromodomain bound to the H3K9me3 histone tail (2RVN) and CHD1 first chromodomain complexed with H3K4me3 (2B2W). Source data are provided with this paper.

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

## Acknowledgements

We thank all the members of the Vannini lab for critically reading the manuscript and for fruitful discussions. We further acknowledge the eBIC Cryo-EM facilities at the UK national electron bio-imaging centre at Diamond, funded by the Wellcome Trust, MRC and BBSRC, for access and support under proposal BI21809. This work was funded by the Cancer Research UK Programme Foundation (CR-UK C47547/A21536) and a Wellcome Trust Investigator Award (200818/Z/16/Z) to A.V.

## Author contributions

G.A.-P. designed and performed the experiments, analysed the data and wrote the manuscript. L.J. produced recombinant TFIIIC. C.P.-P. cloned and carried out initial characterisation of Ty3 integrase. F.B. carried out cryo-EM sample preparation, screening and sample collection and helped with data analysis. A.V. designed and coordinated the project, analysed data and wrote the manuscript.

## Competing interests

The authors declare no competing interests.
