## [Peer Review File · Nature Communications]

Structural basis of Ty3 retrotransposon integration at RNA
Polymerase III-transcribed genesReviewers' Comments:

Reviewer #1:

Remarks to the Author:

In this very interesting work, the authors describe a structure of an intasome from an endogenous retroelement in complex with DNA and its targeting factor (TFIIIB). As such, it substantially extends previous intasome structural studies. First, it presents a structure of an intasome from an ancient endogenous retroelement (and it is very satisfying to see that it shares the CIC with retroviral intasomes, for example). Secondly, it elucidates a mechanism evolved for a precise integration targeting (something that exogenous retroviruses, such as HIV-1, do not require/possess). All of my major comments are related to the description of the intasome structure, which must be improved prior to publication.

Major comments:

1) Structural figures are not very informative. How does Ty3 intasome compare to other tetrameric intasomes, such as PFV or HTLV-I/STLV (6Z2Y, 6VOY)? It is very hard to find the intasome within Figure 2. Figures 5b-d do not show the IN domains; it is impossible to see why Ty3 intasome has a seemingly larger mass. Is this because of ordered loops, linkers, additional ordered domains, or were the structures simply not drawn to scale? It would really help to show the Ty3 intasome side-by-side with PFV: the whole structure (highlight the CIC), as well as an inner subunit isolated from the rest (side-by-side with PFV, for example), even as a supplementary Figure.

2) Which two of the four IN subunits provide the synaptic CTDs and how does the alpha-helical CCD-CTD linker allow positioning of the synaptic CTDs? Do you observe one of both other two CTDs? Does NED share the fold with PFV IN NED? Do you see density for all 4 NTDs/NEDs?

3) When describing the tetrameric intasome assemblies, the catalytic IN chains are called "inner subunits" and the non-catalytic "outer subunits". Description of the more complex retroviral intasomes (octamers, dodecamers, and hexadecamers) requires "peripheral", "distal", etc.

4) Stoichiometry of the HIV-1 intasome is at best ambiguous; and the tetramer (shown in Fig. 5d) is unlikely to represent the native species (as explained in the original paper describing the structure, Pubmed 28059769). Furthermore, even within the CIC tetramer, there are 2 types of possible locations for LEDGF IBD; and there are 8 possible types of IBD locations within a 16-mer (Pubmed 28059770). For these reasons, I suggest toning down the highly speculative "topological" argument (Fig. 5d; bottom of page 9; Abstract).

5) The IN CCD-CTD linker has been discussed extensively in the literature. Along with the CTD, this linker has profound effects on the intasome architecture. Incidentally, in lentiviral INs (HIV-1, SIV, MVV) the CCD-CTD linker is alpha-helical (see detailed analysis in Pubmed 28059770, or a review 34244677; and several published structures). For this reason, it is not correct to contrast Ty3 and HIV-1 based on secondary structure of the linker. Incidentally, the extended CCD-CTD linker of HTLV-I/STLV IN recruits a host factor (6Z2Y, 6VOY).

6) What is the local map resolution range for the intasome part?

Minor comments:

1) The intasome symbols (four overlapping circles plus two DNA sticks) used in Figure 1 and S1 seem to suggest that the outer Ty3 IN chains interact with transposon DNA ends. Is this the case??? In PFV the inner subunits that account for all interactions with viral DNA ends. In HTLV-I/STLV intasome, the outer chains do contact viral DNA ends, but only via CTDs.

2) Page 6: "Careful analysis of the structure reveals that a phosphodiester bond has been formed between the 3'-end of the LTR reactive strands and a cleaved 5'-end on the target DNA, confirming the activity of the assembled complex and suggesting that the majority of the particles imaged by cryo-EM correspond to a strand-transfer complex (Fig. 2b)." I see no evidence of a "phosphodiester bond" in Fig 2b.

3) Related to the previous comment, the authors should define what strand transfer complex is (and how it is different from target capture complex, for example).

4) page 7: "Remarkably, the same region of Ty3 IN, characterized by a compact b-barrel architecture, also mediates tight interactions with the target DNA minor groove when in the context of the core Ty3 subunits, highlighting the plasticity of Ty3 domains in mediating different functions". If anything, this observation highlights plasticity of the CTD function, which is a common theme in intasome structures.

5) The structure nicely explains how Ty3 integration is positioned and stimulated by TFIIB. However, specific integration must also include restriction of catalysis in the absence of TFIIB. Is there evidence for auto-inhibitory functions within retrotransposon INs?

Reviewer #2:

Remarks to the Author:

Abascal-Palacios et al., provide a cryo-EM structure of the Ty3 TFIIB-intasome strand transfer complex. The structure provides key details on how Ty3 is specifically targeted to Pol III transcription start sites. Comparison to other intasome complexes from retroviruses and retrotransposons reveals a conserved architecture between unrelated proteins, suggesting a conserved mechanism of action. The manuscript would benefit from additional biochemical experiments and the inclusion of key experimental details. Specific comments are provided below. Overall, the manuscript will be of broad interest to the transcription community and those studying retrotransposons.

Specific comments

1-The authors claim that the CTD and CHD domains of the Ty3 integrase are required for the interaction with TFIIB and the promoter DNA. They also note that the CHD domain may serve a ruler function. The authors have not provided any biochemical evidence in this manuscript to support these claims. Minimally, the authors should make truncation mutants to test whether these domains reduce or abrogate retrotransposition in their retrotransposon assay. Currently, the authors have cited literature but have not formally shown that the Ty3 integrase requires these domains.

Similarly, the authors claim that an unstructured region of TFIIB could prevent Ty3 association with TFIIB at Pol II promoters. The authors should present biochemical data supporting this claim or tone down the language in this section (page 8 end of first paragraph).

2-The authors should provide some kind of data for their purified proteins, such as an SDS-PAGE of the isolated proteins. Additionally, SDS-PAGE images of the complexes purified by size exclusion chromatography and used for cryo-EM analysis should be provided.

3-A figure of the density used to model the nucleic acids would be helpful. This should probably substitute figure 3 in the main text (current figure 3 is useful as an extended data figure). It would help to understand the geometry of the DNA without proteins being shown. Additionally, it would be helpful to point out where TBP binds and how this results in DNA bending (currently Figure 2b) in a figure panel of the DNA.

4-Cryo-EM data analysis is lacking some key details. The authors should provide masked and

unmasked versions of their final FSC curves, scale bars should be provided for representative micrographs and 2D classes (extended data figure 2), and an angular distribution plot for the final reconstruction. The model in Extended figure panel g appears to be over sharpened. Finally, the authors should provide representative densities for key domains like the CTD and CHD domains of the Ty3 integrase. It would also help to provide representative densities for the DNA and TFIIB.

Minor comments

-Pg. 9 first paragraph- citation(s) is/are needed for the first sentence.

-Pg. 5 last paragraph- It is stated that the authors obtained "a well resolved map". Could the authors comment on resolution and provide an FSC curve for this map in addition to the provided EM densities?

-Pg. 6 "Peripheric" is not a commonly used word. Peripheral would be more suitable here

-Figure 3b-indicate that this is an overlay in the figure itself. Also may help reader to point out specific clashes between the two complexes.

-Figure 5a is not particularly clear. Perhaps it would help to remove the protein that is currently in the background and only superimpose the various linkers and CTD domains?

Reviewer #1 (Remarks to the Author):

In this very interesting work, the authors describe a structure of an intasome from an endogenous retroelement in complex with DNA and its targeting factor (TFIIIB). As such, it substantially extends previous intasome structural studies. First, it presents a structure of an intasome from an ancient endogenous retroelement (and it is very satisfying to see that it shares the CIC with retroviral intasomes, for example). Secondly, it elucidates a mechanism evolved for a precise integration targeting (something that exogenous retroviruses, such as HIV-1, do not require/possess). All of my major comments are related to the description of the intasome structure, which must be improved prior to publication.

Major comments:

1) Structural figures are not very informative. How does Ty3 intasome compare to other tetrameric intasomes, such as PFV or HTLV-I/STLV (6Z2Y, 6VOY)? It is very hard to find the intasome within Figure 2. Figures 5b-d do not show the IN domains; it is impossible to see why Ty3 intasome has a seemingly larger mass. Is this because of ordered loops, linkers, additional ordered domains or were the structures simply not drawn to scale? It would really help to show the Ty3 intasome side-by-side with PFV: the whole structure (highlight the CIC), as well as an inner subunit isolated from the rest (side-by-side with PFV, for example), even as a supplementary Figure.

We are sorry (and agree) that the original structural figures did not clearly show the details about Ty3 intasome organisation. We have now prepared a new figure (Extended Data Figure 4) with a side-by-side comparison between Ty3 (ED Fig. 4a-c) and PFV (ED Fig. 4d-f) intasome architectures. We hope it is clearer now.

As for the difference in molecular mass between Ty3 and PFV, we have included a scaled schematic representation of both integrases and their corresponding molecular weights. Briefly, Ty3 larger mass (in excess of 18 kDa per monomer) arises mainly from the existence of an extended C-terminal region (ca. 11 kDa, which includes the CHD) but also from the presence of longer linkers, which result in slightly larger domains. The reason behind Ty3 intasome (ED Fig. 4a) seemingly larger mass when compared to PFV intasome structural model (ED Fig. 4d) arises from the presence of outer domains NED3/NTD3/CTD3/CHD3 and NED4/NTD4, which were not solved in PFV model (ED Fig. 4d) accounting for approximately 60% extra mass.

2) Which two of the four IN subunits provide the synaptic CTDs and how does the alpha-helical CCD-CTD linker allow positioning of the synaptic CTDs? Do you observe one of both other two CTDs? Does NED share the fold with PFV IN NED? Do you see density for all 4 NTDs/NEDs?

As for comment 1), we hope new Extended Data Fig. 4 provides more details about Ty3 intasome architecture.

In our cryo-EM reconstruction we observe two synaptic CTDs, which are provided by the inner integrase subunits (Extended Data Fig. 4b, c). Ty3 CCD-CTD linker, although longer than in PFV integrase, mediates a similar function in positioning the synaptic CTDs close to the active site (ED Fig 4b, c vs ED Fig 4e, f).

Albeit the cryo-EM density is weaker in some peripheral regions, the general fold of NED and NTD domains seems very similar to that observed in PFV. However, the relative orientation between both domains seems to adopt slightly different paths. Thus, whereas in PFV integrase the inner NED-NTDs seem to follow a "linear" architecture (ED Fig 4f), in Ty3 integrase the inner NED is slightly bent to allow binding to the viral DNA (ED Fig 4c). These changes might be triggered by the binding of 4 Ty3 IN subunits to the LTR ends (instead of the 2 subunits observed in PFV), which leads to a narrower distance between viral DNA molecules when compared to PFV.

3) When describing the tetrameric intasome assemblies, the catalytic IN chains are called "inner subunits" and the non-catalytic "outer subunits". Description of the more complex retroviral intasomes (octamers, dodecamers, and hexadecamers) requires "peripheral", "distal", etc.

Thanks for pointing this out. We have corrected the nomenclature to use the terms "inner" and "outer" when referring to the different integrase subunits.

4) Stoichiometry of the HIV-1 intasome is at best ambiguous; and the tetramer (shown in Fig. 5d) is unlikely to represent the native species (as explained in the original paper describing the structure,

Pubmed 28059769). Furthermore, even within the CIC tetramer, there are 2 types of possible locations for LEDGF IBD; and there are 8 possible types of IBD locations within a 16-mer (Pubmed 28059770). For these reasons, I suggest toning down the highly speculative "topological" argument (Fig. 5d; bottom of page 9; Abstract).

Thanks for highlighting this point. We have now changed the abstract to indicate that Ty3 "...shares similarities with potential LEDGF targeting...". Likewise, we have toned down the sentence in page 9 and repanelled Figure 5 to show the existence of an alternative LEDGF binding site at HIV tetramer. We hope these changes help.

5) The IN CCD-CTD linker has been discussed extensively in the literature. Along with the CTD, this linker has profound effects on the intasome architecture. Incidentally, in lentiviral INs (HIV-1, SIV, MVV) the CCD-CTD linker is alpha-helical (see detailed analysis in Pubmed 28059770, or a review 34244677; and several published structures). For this reason, it is not correct to contrast Ty3 and HIV-1 based on secondary structure of the linker. Incidentally, the extended CCD-CTD linker of HTLV-I/STLV IN recruits a host factor (6Z2Y, 6VOY).

Thank you for pointing this out. We have prepared Extended Data Figure 4 that we hope will provide better understanding of how Ty3 CCD-CTD linker participates in intasome architecture.

Likewise, in page 9 we have removed the comparison of Ty3 and HIV-1 based on the structure of the linker and highlighted that other lentiviral INs CCD-CTD linkers also adopt alpha-helical organisations. Also, to provide additional context we have included a mention in the manuscript and a panel showing the recruitment of HTLV-I (Fig 5f). Finally, with the aim of toning down the text we have stated that "further structural studies will be required to clarify the role of the peripheral regions".

6) What is the local map resolution range for the intasome part?

Thank you for the comment. The resolution of Ty3 intasome ranges from ca. 3-7 Å. We have included a label in Extended Data Figure 2g that points out the general position of TFIIB and Ty3 intasome in the cryo-EM reconstruction coloured by local resolution. TFIIB adopts a "peripheral" position and it is estimated to have lower "average" resolution whereas Ty3 intasome region shows higher resolution.

Minor comments:

1) The intasome symbols (four overlapping circles plus two DNA sticks) used in Figure 1 and S1 seem to suggest that the outer Ty3 IN chains interact with transposon DNA ends. Is this the case??? In PFV the inner subunits that account for all interactions with viral DNA ends. In HTLV-I/STLV intasome, the outer chains do contact viral DNA ends, but only via CTDs.

Thanks for the comment. The schematic representation of Ty3 intasome, TFIIB and TFIIC shown in Figure 1a and S1 was aimed at addressing the "general" binding position to the DNA and not the specific domain/subunit organisation.

As for Ty3 IN binding to the viral DNA ends, the NED and NTD domains of the four Ty3 integrase subunits (both "inner" and "outer") are involved in interactions with the LTRs. We have prepared a new figure (Extended Data Figure 4a) that we hope shows more clearly the involvement of inner and outer NED/NTDs in viral DNA binding.

2) Page 6: "Careful analysis of the structure reveals that a phosphodiester bond has been formed between the 3'-end of the LTR reactive strands and a cleaved 5'-end on the target DNA, confirming the activity of the assembled complex and suggesting that the majority of the particles imaged by cryo-EM correspond to a strand-transfer complex (Fig. 2b)." I see no evidence of a "phosphodiester bond" in Fig 2b.

We thank the reviewer for the comment. We have toned down the language indicating that "...a phosphodiester bond might be formed...". Additionally, we have created an additional panel (ED Fig. 3c) showing detail of the nucleic acid cryo-EM density around the active site and the entire DNA molecule/s.

3) Related to the previous comment, the authors should define what strand transfer complex is (and how it is different from target capture complex, for example).

We have done this in page 6, thank you.

4) page 7: "Remarkably, the same region of Ty3 IN, characterized by a compact b-barrel architecture, also mediates tight interactions with the target DNA minor groove when in the context of the core Ty3 subunits, highlighting the plasticity of Ty3 domains in mediating different functions". If anything, this observation highlights plasticity of the CTD function, which is a common theme in intasome structures.

Thank you for the comment. We have now changed the text in Page 7 to "...highlighting the plasticity of the CTD function as previously reported for other retroelements."

5) The structure nicely explains how Ty3 integration is positioned and stimulated by TFIIIB. However, specific integration must also include restriction of catalysis in the absence of TFIIIB. Is there evidence for auto-inhibitory functions within retrotransposon INs?

This is an interesting open question. Although we have not found evidence of any auto-inhibitory mechanism in the complex reconstituted *in-vitro* we cannot discard the existence of such function. The solved structure only shows the presence of 2 chromodomains, both of them apparently involved in the process of integration. However, the two remaining CHDs might be involved in restriction of the activity that could only be observed in the absence of targeting factors. This is a very interesting point and further structural studies will be needed to clarify it.

Reviewer #2 (Remarks to the Author):

Abascal-Palacios et al., provide a cryo-EM structure of the Ty3 TFIIIB-intasome strand transfer complex. The structure provides key details on how Ty3 is specifically targeted to Pol III transcription start sites. Comparison to other intasome complexes from retroviruses and retrotransposons reveals a conserved architecture between unrelated proteins, suggesting a conserved mechanism of action. The manuscript would benefit from additional biochemical experiments and the inclusion of key experimental details. Specific comments are provided below. Overall, the manuscript will be of broad interest to the transcription community and those studying retrotransposons.

Specific comments

1-The authors claim that the CTD and CHD domains of the Ty3 integrase are required for the interaction with TFIIIB and the promoter DNA. They also note that the CHD domain may serve a ruler function. The authors have not provided any biochemical evidence in this manuscript to support these claims. Minimally, the authors should make truncation mutants to test whether these domains reduce or abrogate retrotransposition in their retrotransposon assay. Currently, the authors have cited literature but have not formally shown that the Ty3 integrase requires these domains.

Thank you for the comment. Unfortunately, given the relevance of Ty3 integrase C-terminal domain (CTD) in the stabilisation of the integration active site (through the intasome inner subunits), truncations of this domain would cause a direct disruption of the activity that could not be discerned from effects in the recruitment.

Regarding the chromodomain function, we have cloned and purified a truncation mutant of Ty3 integrase (aa. 1-434) that lacks this domain. In spite of the high protein purity (see included SDS-PAGE), our attempts to reconstitute the Ty3 intasome using this construct proved to be unsuccessful. We believe this highlights the complementary function of Ty3 CHD in steps associated with integrase tetramer formation, before targeting of the host factors. Indeed, the plasticity of integrase domains is a known subject in intasome structures. Moreover, in our structure we only observed 2 of the 4 chromodomains present in the complex (those involved in interactions with TFIIIB and the DNA) and we think that the missing domains could adopt a free conformation in the reconstituted complex and participate in alternative functions. In fact, charged-to-alanine scanning mutagenesis of key chromodomain residues has been performed in the past (Nymark-McMahon, M. H. and S. B. Sandmeyer (1999). *J Virol* 73(1): 453-465) and lead to IN instability, defects in preintegration complex formation and abrogation of retrotransposition:

- **"... mutations in the carboxyl-terminal domain [431A(2), 442A(2), and 496A(2)] resulted in undetectable levels of mature IN and two [477A(2) and 488A(2)]**

displayed a species that was smaller than 61 kDa and also present at lower levels than wild-type IN. Thus, mutations in nonconserved regions of Ty3 IN can affect IN processing and/or stability.

- *“... can be argued that mutants that lack IN completely, in particular 442A(2) and 496A(2), may be unable to form stable preintegration complexes.”*
- *“The phenotypes of the IN mutants in this study are complex and suggest that IN participates in multiple aspects of the Ty3 life cycle”*

In summary, considering the versatility of other IN domains (i.e. CTD role in host targeting and in the active site) and the known DNA- and protein-binding abilities of chromodomains, we believe that Ty3 CHDs could be involved in earlier steps of intasome formation as well as in TFIIIB targeting.

Similarly, the authors claim that an unstructured region of TFIIIB could prevent Ty3 association with TFIIIB at Pol II promoters. The authors should present biochemical data supporting this claim or tone down the language in this section (page 8 end of first paragraph).

Thanks to the reviewer for the helpful comment. As it would be difficult to prove this point experimentally, as suggested, we have toned down the language.

2-The authors should provide some kind of data for their purified proteins, such as an SDS-PAGE of the isolated proteins. Additionally, SDS-PAGE images of the complexes purified by size exclusion chromatography and used for cryo-EM analysis should be provided.

Thank you, we appreciate the reviewer comments. In the revised version of the manuscript we have included SDS-PAGEs of the purified proteins (Extended Data Fig. 1a) and of the purified complexes after gel-filtration chromatography (Extended Data Fig. 1d, e).

3-A figure of the density used to model the nucleic acids would be helpful. This should probably substitute figure 3 in the main text (current figure 3 is useful as an extended data figure). It would help to understand the geometry of the DNA without proteins being shown. Additionally, it would be helpful to point out where TBP binds and how this results in DNA bending (currently Figure 2b) in a figure panel of the DNA.

We have created the new Extended Data Fig. 3, which includes detail of the density used for nucleic acid building and of the DNA bending, TBP binding position, etc. (panel c). Thank you for the point, we think the figure looks much better now and we hope it helps with data interpretation. Considering ED Fig 3 is an extensive figure which provides not only information about DNA but also about Ty3 integrase and transcription factors density, we believe it fits better as a supplementary figure.

4-Cryo-EM data analysis is lacking some key details. The authors should provide masked and unmasked versions of their final FSC curves, scale bars should be provided for representative micrographs and 2D classes (extended data figure 2), and an angular distribution plot for the final reconstruction. The model in Extended figure panel g appears to be over sharpened. Finally, the authors should provide representative densities for key domains like the CTD and CHD domains of the Ty3 integrase. It would also help to provide representative densities for the DNA and TFIIIB.

Thanks for spotting this. The revised version of Extended Data Figure 2 now includes the masked and unmasked versions of the final FSC curves (panel e), the scale bars for the micrographs and 2D classes and an angular distribution plot for the final reconstruction (panel f).

A new figure (Extended Data Figure 3) has been created which shows details about the cryo-EM reconstruction and model fitting, including a correctly sharpened map (panel a) and representative densities for Ty3 integrase domains, TFIIIB and the DNA (panels b-e).

Minor comments

-Pg. 9 first paragraph- citation(s) is/are needed for the first sentence.

Thank you for pointing this out. We have now added the corresponding references.

-Pg. 5 last paragraph- It is stated that the authors obtained “a well resolved map”. Could the authors comment on resolution and provide an FSC curve for this map in addition to the provided EM densities?

Apologies for the confusion. The map in Extended Data Fig. 1h corresponds to an *ab-initio* reconstruction and it was used as a very preliminary approach to test the fit of the different components of the complex before further characterisation by cryoEM. Further refinement or assessment of its resolution was not performed and therefore the use of the term “well resolved” is incorrect. We have now removed it from the text. Thank you.

-Pg. 6 “Peripheric” is not a commonly used word. Peripheral would be more suitable here
We have corrected this typo, thank you.

-Figure 3b-indicate that this is an overlay in the figure itself. Also may help reader to point out specific clashes between the two complexes.

We have changed the way NC2 β is depicted in Figure 4b using a molecular surface representation. We think the new panel shows more clearly the potential clashes between NC2 β and Ty3 integrase. Likewise, to indicate that this panel corresponds to an overlay, the figure legend now reads “*Structural models are superimposed on TBP molecule*”.

-Figure 5a is not particularly clear. Perhaps it would help to remove the protein that is currently in the background and only superimpose the various linkers and CTD domains?

Thank you, we appreciate that the original figure did not clearly show the described differences. We have prepared a new Figure 5, where the retroelements have been depicted individually and the background molecules have been made transparent. Likewise, the nucleic acids have been included as a reference of the “inner” or “outer” orientation of the C-terminal domains (CTD).

Reviewers' Comments:

Reviewer #1:

Remarks to the Author:

Most of my concerns have been addressed, and the manuscript improved in revision. However, unless the authors can show (i) a convincing superposition of LEDGF and Brf1 structures and (ii) determine where exactly on HIV-1 intasome LEDGF binds, they should remove "Surprisingly, Ty3 tethering shares similarities to with LEDGF..." from their Abstract (lines 35-38).

Additionally:

Line 23: "integrating into" (rather than onto).

Line 61: "HIV binds to the transcriptional co-activator....". Surely this can be written better (HIV uses...; integrase binds...; ...).

Line 198: strand transfer itself does not generate a TSD. Formation of a TSD requires DNA synthesis. This part will be very confusing for anyone who does not know much about integration.

Reviewer #2:

Remarks to the Author:

The authors have addressed my concerns. Reviewers' Comments:

Reviewer #1:

Remarks to the Author:

Most of my concerns have been addressed, and the manuscript improved in revision. However, unless the authors can show (i) a convincing superposition of LEDGF and Brf1 structures and (ii) determine where exactly on HIV-1 intasome LEDGF binds, they should remove "Surprisingly, Ty3 tethering shares similarities to with LEDGF..." from their Abstract (lines 35-38).

Additionally:

Line 23: "integrating into" (rather than onto).

Line 61: "HIV binds to the transcriptional co-activator....". Surely this can be written better (HIV uses...; integrase binds...; ...).

Line 198: strand transfer itself does not generate a TSD. Formation of a TSD requires DNA synthesis. This part will be very confusing for anyone who does not know much about integration.

Reviewer #2:

Remarks to the Author:

The authors have addressed my concerns.

We would like thank the reviewers again for the time and effort invested in reviewing our manuscript “Structural basis of Ty3 retrotransposon integration at RNA Polymerase III-transcribed genes” by Guillermo Abascal-Palacios, Laura Jochem, Carlos Pla-Prats, Fabienne Beuron and Alessandro Vannini. The manuscript has significantly improved thanks to their very useful comments. Here below, please find a point-by-point respond to their comments.

Reviewer #1 (Remarks to the Author):

Most of my concerns have been addressed, and the manuscript improved in revision. However, unless the authors can show (i) a convincing superposition of LEDGF and Brf1 structures and (ii) determine where exactly on HIV-1 intasome LEDGF binds, they should remove "Surprisingly, Ty3 tethering shares similarities to with LEDGF..." from their Abstract (lines 35-38).

Thanks for the comment. We have removed this sentence as part of the Abstract remodelling to fit into 250 words.

Additionally:

Line 23: "integrating into" (rather than onto).

Thanks for pointing this out. We have now corrected this typo.

Line 61: "HIV binds to the transcriptional co-activator....". Surely this can be written better (HIV uses...; integrase binds...; ...).

Thank you. We have changed the sentence to “...(HIV) integrase binds to the...”.

Line 198: strand transfer itself does not generate a TSD. Formation of a TSD requires DNA synthesis. This part will be very confusing for anyone who does not know much about integration.

Thank you. We have modified the sentence into “...which would result in a 5-bp target site duplication (TSD) after a final step of DNA synthesis”. We hope it is clearer now.

Reviewer #2 (Remarks to the Author):

The authors have addressed my concerns.